# Genome-Wide Identification and Characterization of WRKY Transcription Factors in *Betula platyphylla* Suk. and Their Responses to Abiotic Stresses

**DOI:** 10.3390/ijms241915000

**Published:** 2023-10-08

**Authors:** Jiajie Yu, Xiang Zhang, Jiayu Cao, Heming Bai, Ruiqi Wang, Chao Wang, Zhiru Xu, Chunming Li, Guanjun Liu

**Affiliations:** 1State Key Laboratory of Tree Genetics and Breeding, Northeast Forestry University, Harbin 150040, China; 13514554189@163.com (J.Y.); nefuzhangxiang@163.com (X.Z.); 13766664352@163.com (J.C.); bai199910@163.com (H.B.); yjswrq@outlook.com (R.W.); wangchao@nefu.edu.cn (C.W.); xuzhiru2003@126.com (Z.X.); 2College of Life Science, Northeast Forestry University, Harbin 150040, China

**Keywords:** *Betula platyphylla*, WRKY transcription factor, bioinformatics analysis, expression pattern, ABA treatment, salt treatment, cold treatment, protein interaction, subcellular localization

## Abstract

The WRKY transcription factor (TF) family is one the largest plant-specific transcription factor families. It has been proven to play significant roles in multiple plant biological processes, especially stress response. Although many WRKY TFs have been identified in various plant species, WRKYs in white birch (*Betula platyphylla* Suk.) remain to be studied. Here, we identified a total of 68 BpWRKYs, which could be classified into four main groups. The basic physiochemical properties of these TFs were analyzed using bioinformatics tools, including molecular weight, isoelectric point, chromosome location, and predicted subcellular localization. Most BpWRKYs were predicted to be located in the nucleus. Synteny analysis found 17 syntenic gene pairs among *BpWRKY*s and 52 syntenic gene pairs between *BpWRKY*s and *AtWRKY*s. The *cis*-acting elements in the promoters of *BpWRKY*s could be enriched in multiple plant biological processes, including stress response, hormone response, growth and development, and binding sites. Tissue-specific expression analysis using qRT-PCR showed that most *BpWRKY*s exhibited highest expression levels in the root. After ABA, salt (NaCl), or cold treatment, different *BpWRKY*s showed different expression patterns at different treatment times. Furthermore, the results of the Y2H assay proved the interaction between BpWRKY17 and a cold-responsive TF, BpCBF7. By transient expression assay, BpWRKY17 and BpWRKY67 were localized in the nucleus, consistent with the previous prediction. Our study hopes to shed light for research on WRKY TFs and plant stress response.

## 1. Introduction

Unable to relocate, plants have to cope with a variety of adverse environmental conditions. In the short-term or long-term, abiotic stresses like drought, salt, and temperature stress greatly influence the growth and development of plants and even pose a threat to crop yield and food security. Plants have evolved intricate mechanisms to avoid or mitigate the damage caused by abiotic stress, one of which is transcriptional regulation [1]. Transcription factors are the core module in this process. They can activate or repress the expression of downstream genes by binding to the *cis*-elements in their promoters [2]. The WRKY transcription factor (TF) superfamily, characterized by the conserved WRKY domain, is one of the largest plant-specific transcription factor families [3]. The members in this family contain two structural parts, the N-terminal DNA binding domain and the C-terminal zinc-finger motif (C-X_4–5_-C-X_22–23_-HXH (C_2_H_2_) or C-X_7_-C-X_23_-HXC (C_2_HC)) [4]. The N-terminal DNA-binding domain is characterized by a WRKYGQK motif, of which several variants exist, including WRKYGKK, WRKYGRK, FWRKYGQK, WRKYGEK, WRKYGNK, and WRKYGHK [5,6]. Otherwise, there are also variants of the WRKY sequence, such as WKKY, WSKY, WKRY, WIKY, WVKY, WRIC, and WRMC [7]. According to the number of WRKY motifs and the characteristics of the zinc-finger motif, WRKY TFs can be divided into four groups (Ⅰ–IV) [8,9]. Group Ⅰ possesses two WRKY motifs; Group Ⅱ possesses one WRKY motif and a C_2_H_2_ motif; Group III possesses one WRKY motif and a C_2_HC motif; and Group IV possesses an incomplete WRKY motif and no zinc-finger motifs.

The first member (*SPF1*) of the WRKY TF family was identified in potato (*Impoea batatas*) in 1994 [10]. With the development of genome sequencing, more genome information of various plant species has been unveiled and become available for the identification of TF family members through the whole genome. To date, WRKY TFs have been identified in many plant species, including 72 in *Arabidopsis thaliana* [11], 56 in *Camellia sinensis* [12], 48 in *Panax ginseng* [13], 103 in *Oryza sativa* [14], 119 in *Zea mays* [15], 79 in *Eucalyptus grandis* [16], 119 in *Malus domestica* [17], 107 in *Populus euphratica* [18], and 98 in *Populus trichocarpa* [19]. To date, an increasing body of evidence has proven the important roles of WRKY TFs involved in multiple plant biological processes, including hormone signal transduction [20,21,22], seed germination [23], senescence [24], and especially, stress response [25]. To cope with the biotic stress response, plants rely on related transcriptional regulation, in which WRKY TFs have been extensively studied and proven to be closely involved. In Sun’s study, the *SlWRKY39* gene was induced by salt, drought stress, or PstDC3000 infection [26]. Overexpression of this gene in tomato improved the resistance of transgenic plants against PstDC3000 infection, as well as their tolerance to drought and salt stress. In rice, overexpression of *OsWRKY62.1* compromised the basal defense and *Xa21*-mediated resistance against the bacterial pathogen *Xanthomonas oryzae* pv. *oryzae* [27]. A large number of studies have also proven that WRKY TFs play significant roles in plant responses to abiotic stress. In *Arabidopsis*, *AtWRKY25* and *AtWRKY33* were induced by salt (NaCl) stress [28]. Further qRT-PCR results showed that *AtWRKY25* was responsive to NaCl, mannitol, ABA, and cold (4 °C) treatment. *AtWRKY33* was responsive to NaCl, mannitol, and cold treatment. Overexpression of *AtWRKY25* or *AtWRKY33* in *Arabidopsis* increased the salt tolerance of transgenic plants. *GmWRKY16*, a WRKY TF in soybean, was responsive to salt, alkali, ABA, drought, or PEG-6000 treatment [29]. *Arabidopsis* plants overexpressing this gene showed stronger tolerance to salt and drought stress through an ABA-mediated pathway. In sugar beet, *BvWRKY3*, -*10*, -*16*, -*22*, -*41*, -*42*, -*44*, -*47*, and -*51* were induced by different concentrations of NaHCO_3_, which indicated the important roles of these genes in plant response to alkaline stress [30]. In maize, the expression of *ZmWRKY106* was significantly upregulated by drought, high temperature, and ABA treatment [31]. Overexpression of this gene in *Arabidopsis* boosted the tolerance of transgenic plants to drought and high temperature stress. *PmWRKY57*, a member of WRKY TFs in *Prunus mume*, was induced by cold stress [32]. Overexpression of this gene in *Arabidopsis* improved the tolerance of transgenic plants to cold stress. In durum wheat, alterations of two amino acids close to the WRKY domain resulted in a functional modification in the biochemistry of the *WRKY1* promoter, which contributed to the salt resistance of two durum wheat genotypes [33]. In tomato, the transcription level of the *SlWRKY81* gene was gradually increased upon drought stress [34]. Silencing of this gene in tomato enhanced the tolerance of transgenic plants to drought stress, and overexpression of this gene generated an opposite phenotype.

White birch (*Betula platyphylla* Suk.) is widely distributed across the globe, especially in subarctic and temperate Asia and in areas of countries like Russia, Mongolia, China, Japan, and Korea. Economically, white birch can be used for the production of several kinds of foods and medicines, as well as for paper and architecture. Ecologically, white birch is of great importance to soil improvement, water conservation, and sandstorm control. It is a well-known and prevalent research material because of its excellent performance in terms of adaptation, tolerance, and resistance against biotic and abiotic stresses [35]. However, WRKY TFs have rarely been identified or studied in birch, and their responses to abiotic stresses remain undetermined.

Therefore, this study aims to identify WRKY transcription factors at a whole-genome scale in birch and investigate their responses to abiotic stresses. In this study, we identified a total of 68 WRKY TF family members in *Betula platyphylla*. Their basic physiochemical properties, phylogenetic relationships, and gene structures were investigated. Synteny analysis was performed on the WRKYs in *Betula platyphylla* as well as in *Betula platyphylla* and *Arabidopsis thaliana*. After tissue-specific analysis and *cis*-acting element analysis, the responses of *BpWRKY*s upon ABA, salt (NaCl), and cold treatment were investigated by qRT-PCR. We also illustrated the interaction between BpWRKY17 and a cold-responsive TF, BpCBF7, and verified the subcellular localization of BpWRKY17 and BpWRKY67. This study hopes to shed light for future research on the mechanisms of BpWRKYs in plant stress response.

## 2. Results

### 2.1. Identification and Characterization of WRKY Transcription Factors in Betula platyphylla

To explore all of the potential WRKY transcription factors in *Betula platyphylla*, genome data downloaded from the phytozome were searched with BlastP, and the hidden Markov model (PF03106) was used to confirm the existence of the WRKY domain. Then, through InterPro and CDD screening, a total of 68 *BpWRKY* genes were identified and designated as *BpWRKY1* to *BpWRKY68* according to their location order on chromosomes. The 68 *BpWRKY*s were distributed on 14 chromosomes and scaffolds (Figure 1). On Chromosomes 5 and 11 lay the most BpWRKYs, with 10 members on each chromosome. On Chromosomes 3 and 12 lay the least *BpWRKY*s, with only one member on each chromosome. Additionally, *BpWRKY65*, -*66*, -*67*, and -*68* were located on two scaffolds.

To further study the basic characteristics of these TFs, physiochemical properties such as molecular weight, isoelectric point, chromosome location, and predicted cellular localization were investigated and are summarized in Appendix A. The length of BpWRKY proteins ranged from 133 to 752 amino acids. BpWRKY41 was the only TF with a minus GRAVY score, which indicated that only this TF was hydrophobic while the other 67 BpWRKYs were hydrophilic. As for cellular location, most of the BpWRKYs were predicted to be located in nucleus. There were also exceptions in that BpWRKY41 was predicted to be located in the chloroplast and nucleus and BpWRKY47 was located in the cell membrane and nucleus.

### 2.2. Phylogenetic Analysis of BpWRKYs

To accurately investigate the evolutionary relationship between the members of the WRKY TF family in *Betula platyphylla*, WRKYs in *Arabidopsis thaliana* were introduced to construct the phylogenetic tree after multiple sequence alignment (Figure 2). This phylogenetic tree contained a total of 72 AtWRKYs and 68 BpWRKYs and they were classified into four groups (Classes Ⅰ, Ⅱ, III, and IV) according to the traits of the conserved domains. Class Ⅰ included 10 members, which contained two conserved WRKY domains. Class Ⅱ included 42 members, which were further classified three subgroups (Ⅱa, Ⅱb, and Ⅱc). In this class, all members possessed a complete WRKYGQK domain except BpWRKY47, which contained a WRIHGIK domain, and BpWRKY20, -21, and -60, which contained a WRKYGKK domain. Class III included six members, which contained a WRKY domain and a zinc-cluster domain (PF10533). Class IV could be further divided into two subgroups (IVa and IVb) and included nine members, which did not contain an orthodox WRKY domain. For example, BpWRKY41 contained a mutant WRKY domain, WRKYGMK. The results of the multiple sequence alignment are shown in Appendix A.

### 2.3. Gene Structure of BpWRKYs

To better understand the gene structure of *BpWRKY* genes based on the classification, the distributions of exon–intron and conserved motifs were analyzed and visualized using the online tools GSDS 2.0 and MEME. The results were displayed along with a phylogenetic tree of *BpWRKY*s in order to further explore the conservatism and evolutionary relationships between these genes (Figure 3). These classification results were consistent with the results in Section 2.2. Members in the same subgroup exhibited similar characteristics in gene structure. Only the members of Class Ⅰ possessed Motif 3. All of the members of Class Ⅱc possessed Motif 6. Only the members of Class Ⅱb possessed Motif 10. Only the members of Class IVa possessed Motifs 4 and 8. The detailed information of the 10 conserved motifs is listed in Appendix A.

The result of the exon–intron analysis showed that only a quarter of the *BpWRKY*s did not have a 5′-untranslated region (5′-UTR) or 3′-untranslated region (3′-UTR). Approximately a third of the *BpWRKY*s possessed only 5′-UTR without 3′-UTR. Only *BpWRKY9* and *BpWRKY49* possessed only 3′-UTR without 5′-UTR. The 68 *BpWRKY*s all possessed exons and introns. The number of exons ranged from two to seven, while four to five were more frequently seen. The number of introns ranged from one to six.

### 2.4. Synteny Analysis of BpWRKYs

Synteny analysis is a powerful and effective tool for the investigation of gene duplication events. In this case, the syntenic relationships among *BpWRKY*s as well as between *WRKY*s in *Arabidopsis thaliana* and *Betula platyphylla* were analyzed and visualized. The results of the synteny analysis of *WRKY* genes in *Betula platyphylla* are shown in Figure 4. Due to their locations in scaffolds and lack of syntenic relationships, *BpWRKY65*, -*66*, -*67,* and -*68* are not shown here. A total of 17 syntenic gene pairs (Table 1) were found. Ka symbolizes synonymous mutation and Ks symbolizes non-synonymous mutation. Apart from *BpWRKY19*/*BpWRKY22*, the Ka/Ks ratios were all less than 1, which was a clear implication of purifying selection. This exception arose from the complete embrace of the gene sequence of *BpWRKY19* in *BpWRKY22* (Appendix A), which meant the Ka and Ks values of this pair of genes were both 0. This indicated the insertion or deletion of gene segments during the evolution process.

To further investigate the evolutionary relationships between *BpWRKY*s, synteny analysis of *WRKY* genes in *Betula platyphylla* and *Arabidopsis thaliana* was performed (Figure 5). The results showed that there were 52 syntenic gene pairs between *BpWRKY*s and *AtWRKY*s. These syntenic *BpWRKY*s were distributed on all of the chromosomes except Chromosome 9. On Chromosome 11 lay the most syntenic *BpWRKY*s. Some *BpWRKY*s had syntenic relationships with more than one *AtWRKY*. For instance, on Chromosome 7, *BpWRKY33* had syntenic relationships with *AtWRKY7*, -*42*, and -*46*. The gene IDs of the *AtWRKY*s are listed in Appendix A. The syntenic gene pairs are listed in Appendix A.

### 2.5. Analysis of Cis-Acting Elements in the Promoters of BpWRKYs

The *cis*-acting elements in the promoters of *BpWRKY*s were identified and analyzed using PlantCARE. These elements (Figure 6) were mainly related to stress response (such as drought responsive, wound-responsive, antioxidative responsive, and low-temperature responsive elements), hormone response (abscisic acid-responsive, methyl jasmonate-responsive, and gibberellin-responsive elements), growth and development (such as meristem expression and endosperm expression), and binding sites. The most frequently shown *cis*-acting elements included light-responsive and abscisic acid-responsive elements, reaching 286 and 257 elements, respectively. In addition, elements related to stress response had a very large proportion. They were distributed widely and massively in the promoters of almost all *BpWRKY*s, which indicated the potential key roles of *BpWRKY*s in plant stress response. A detailed summary of the *cis*-acting elements is listed in Appendix A.

### 2.6. Expression Pattern Analysis of BpWRKYs

To explore the expression pattern of *BpWRKY*s in different tissues of birch, the RNA of birch roots, stems, and leaves was extracted and then reversely transcribed into cDNA for qRT-PCR. As shown in Figure 7, most *BpWRKY*s were more expressed in the root than in the stem or leaf. This comparison exhibited different degrees. For instance, the expression level of *BpWRKY12* in the root was approximately 15 times that in the stem and 42 times that in the leaf. The expression level of *BpWRKY40* in the root was approximately 8.5 times that in the stem and 11 times that in the leaf. The expression level of BpWRKY26 in the root was approximately 3.4 times that in the stem and 3821 times that in leaf. However, there were also exceptions in that some *BpWRKY*s were most expressed in the stem or leaf. The expression level of *BpWRKY10* in the leaf was approximately 2.8 times that in the stem and 3 times that in the root. The expression level of *BpWRKY29* in the stem was approximately 2.1 times that in the leaf and 14.6 times that in root. In the root, *BpWRKY32* had the highest expression level and *BpWRKY29* had the lowest (log_2_ value ranging from 5.447 to −12.517). In the stem, *BpWRKY24* had the highest expression level and *BpWRKY47* had the lowest (log_2_ value ranging from 4.127 to −13.707). In the leaf, *BpWRKY24* had the highest expression level and *BpWRKY13* had the lowest (log_2_ value ranging from 4.253 to −12.610).The related qRT-PCR data are listed in Appendix A.

### 2.7. Transcriptional Responses of BpWRKYs upon ABA Treatment

The results in Section 2.5 showed an enrichment of stress-responsive *cis*-acting elements in the promoters of *BpWRKY*s, which indicated that these genes could be responsive to abiotic stresses such as ABA, salt, or cold treatment. In this case, the relative expression levels of *BpWRKY*s upon different times (0 h, 3 h, 6 h, 12 h, and 24 h) of ABA treatment were measured using qRT-PCR (Figure 8). The results showed that almost all of the *BpWRKY*s were responsive to ABA treatment while in different expression patterns and to different degrees. Over half of the *BpWRKY*s were downregulated with prolonged time of ABA treatment. The lowest relative expression levels usually showed at the 6 h or 12 h time point. For instance, the expression level of *BpWRKY31* at 0 h was approximately 6.7 times that at 3 h, 3.4 times that at 6 h, 58.9 times that at 12 h, and 2.4 times that at 24 h. At the same time, the expression levels of some *BpWRKY*s were increased with prolonged time of ABA treatment. The expression level of *BpWRKY52* at 0 h was approximately 0.75 times that at 3 h, 0.82 times that at 6 h, 0.5 times that at 12 h, and 1.33 times that at 24 h. In addition, some *BpWRKY*s were not that responsive to ABA treatment, such as *BpWRKY47* and *BpWRKY64*, which kept being expressed at a low level at any time point. The related qRT-PCR data are listed in Appendix A.

### 2.8. Transcriptional Responses of BpWRKYs upon Salt Treatment

To elucidate the expression pattern of *BpWRKY*s upon salt (NaCl) treatment, the relative expression levels of *BpWRKY*s upon different times (0 h, 3 h, 6 h, 12 h, and 24 h) of ABA treatment were measured using qRT-PCR (Figure 9). Similar to the responses to ABA treatment, over half of the *BpWRKY*s were downregulated with prolonged time of salt treatment. For instance, the expression level of *BpWRKY14* at 0 h was approximately 1.25 times that at 3 h, 2.43 times that at 6 h, 2.33 times that at 12 h, and 2.36 times that at 24 h. The expression level of BpWRKY57 at 0 h was approximately 15 times that at 3 h, 28 times that at 6 h, 57 times that at 12 h, and 54 times that at 24 h. There was also a large proportion of *BpWRKY*s that were upregulated upon salt treatment. The expression level of *BpWRKY7* at 0 h was approximately 0.27 times that at 3 h, 0.43 times that at 6 h, 0.39 times that at 12 h, and 0.66 times that at 24 h. The expression level of *BpWRK61* at 0 h was approximately 0.13 time that at 3 h, 0.19 times that at 6 h, 0.16 times that at 12 h, and 0.07 times that at 24 h. Similar to upon ABA treatment, the lowest or highest expression levels of different *BpWRKY*s upon salt treatment occurred at different time points. The related qRT-PCR data are listed in Appendix A.

### 2.9. Transcriptional Responses of BpWRKYs upon Cold Treatment

Because of the wide distribution of low-temperature responsive elements in the promoters of *BpWRKY*s, the responses of these genes upon cold treatment (4 °C for 3 h) were also investigated through qRT-PCR (Figure 10). The results showed that a large quantity of *BpWRKY*s were responsive to cold treatment, although not as many as upon ABA or salt treatment. The expression levels of cold-responsive *BpWRKY*s were increased after cold treatment to different degrees. For instance, the relative expression level of *BpWRKY4* at 3 h was approximately 3.7 times that at 0 h. The relative expression level of *BpWRKY33* at 3 h was approximately 4 times that at 0 h. The relative expression level of *BpWRKY61* at 3 h was approximately 8.9 times that at 0 h. However, there were also some *BpWRKY*s for which the relative expression levels were decreased upon cold treatment. For instance, the relative expression level of *BpWRKY56* at 0 h was approximately 3.7 times that at 3 h. The relative expression level of *BpWRKY6* at 0 h was approximately 7.8 times that at 3 h. The related qRT-PCR data are listed in Appendix A.

### 2.10. Interaction of BpWRKY17 with the Cold-Responsive TF BpCBF7

A previous study of ours demonstrated that the transcriptional activating activity of BpCBF7, a member of the C-repeat binding factor transcription factor subfamily, was responsive to cold treatment [36]. In this study, the transcriptional activating activity of BpCBF7 was removed by cloning the partial *BpCBF7* gene (1–438 bp), and then it was introduced into the pGBKT7 vector for Y2H assay. At the same time, the *BpWRKY17* gene was introduced into the pGADT7 vector. The two recombinant vectors were simultaneously transformed into yeast cells. As shown in Figure 11, the negative control was not able to grow on the nutrition-deprived medium, while the positive control and yeast transformed with pGADT7-BpWRKY17 and pGBKT7-BpCBF7 (1–146 aa) could grow normally, which indicated that BpWRKY17 could interact with BpCBF7. As shown in Section 2.9, *BpWRKY17* was responsive to cold treatment. All of the results showed that BpWRKY17 possibly plays important roles in the response of birch to cold stress, by itself or interaction.

### 2.11. Subcellular Localization of BpWRKY17 and BpWRKY67

To further identify where BpWRKY17 and BpWRKY67 resided in the cell, a transient expression assay was performed by introducing these two genes into the 35S-eGFP vector and transforming the recombinant vectors into tobacco cells. The results (Figure 12) showed that the fluorescence of GFP, as the positive control, could be detected throughout the whole cell, while that of 35S-eGFP-BpWRKY17 or 35S-eGFP-BpWRKY67 could only be detected in the nucleus. These results were consistent with the prediction results in Section 2.1.

## 3. Discussion

Living in the ever-changing environment, plants have to cope with a variety of adverse stresses, such as drought, salt, high or low temperature, and heavy metal stress. In the long period of evolution, transcriptional regulation has become one of the most important strategies employed by plants to tolerate or resist stresses. The WRKY TF family, one of the largest plant-specific TF families, has been proven to be involved in multiple stress responses, especially abiotic stress response [3]. To date, WRKY family members have been identified in many species, such as *Arabidopsis thaliana*, *Camellia japonica*, *Oryza sativa*, *Eucalyptus grandis*, and *Populus trichocarpa* [11,12,14,16,19]. However, WRKY TFs have yet to be identified and characterized in white birch (*Betula platyphylla*), which is widely distributed and has good tolerance to abiotic stress.

In this study, a total of 68 WRKY TFs were identified in *Betula platyphylla*, named BpWRKY1–68. The distribution of *BpWRKY* genes on chromosomes was not so even. For instance, on Chromosome 5 laid ten *BpWRKY*s, while on Chromosome 3 laid only one. There were also four *BpWRKY*s, *BpWRKY65*–*68*, located on scaffolds. The basic physiochemical properties of BpWRKYs were investigated, such as molecular weight, isoelectric point, and predicted subcellular localization. Most of them were predicted to be localized in the nucleus. This is consistent with the fact that more TFs are commonly located in nucleus. There were also exceptions in that BpWRKY41 was predicted to be localized in the chloroplast and nucleus and BpWRKY47 was predicted to be localized in the cell membrane and nucleus. To verify and explore whether the chromosome location would influence the prediction, we performed subcellular localization on BpWRKY17 and BpWRKY67, representatives of BpWRKY of which the coding genes were located on the chromosome or scaffold, respectively. They were both localized in the nucleus, which was consistent with the result of the prediction. This finding indicates that the chromosome location of *BpWRKY*s possibly does not influence the localization of the encoding proteins. The phylogenetic analysis classified the 68 BpWRKYs into four main groups according to the number and characteristics of their WRKY and zinc-finger motifs. In the phylogenetic tree, AtCBFs were also incorporated for better demonstration of the evolutionary relationships. Class Ⅱ could be further classified into three subgroups, Classes Ⅱa, Ⅱb, and Ⅱc, and Class IV could be further classified into two subgroups, Classes IVa and IVb. This resulted from the similarity of the conserved domains. In addition, the members of the same group had similar exon–intron patterns, which indicates conservatism in the evolution process. Synteny analysis can help investigate gene duplication events. In this study, there were 17 syntenic gene pairs found among *BpWRKY*s and 52 syntenic gene pairs found between *BpWRKY*s and *AtWRKY*s. There were 14 in 17 syntenic gene pairs in *BpWRKY*s, which indicates strong purification selection in the evolution process. The disadvantageous non-synonymous substitution gradually disappeared. Gene duplication could be the main reason of the expansion of *BpWRKY*s. Two syntenic gene pairs showed specificity, *BpWRKY19*/*BpWRKY22* and *BpWRKY20*/*BpWRKY21*. The sequence of *BpWRKY22* contained the whole sequence of *BpWRKY19*, which indicated the insertion or deletion of gene segments during the evolution process. The sequences of *BpWRKY20* and *BpWRKY21* exhibited extremely high similarity, which possibly resulted from mutation in gene duplication events.

The tissue-specific analysis showed that most *BpWRKY*s were most expressed in the root, which was consistent with some previous studies on WRKY TFs in other species [37,38]. This indicates that WRKY TFs show a degree of similarity in tissue-specific expression across some different plant species. The *cis*-acting elements in the promoters of *BpWRKY*s were mainly enriched in stress response, hormone response, growth and development, and binding sites. This finding indicates that *BpWRKY*s may be responsive to abiotic stress and play important roles in hormone signaling. For instance, in *Glycine max*, *GmWRKY13* is involved in the salt stress response through the ABA signaling pathway. Overexpression of this gene in *Arabidopsis* decreased the salt tolerance of transgenic plants by regulating *ABI1*, a negative regulator in the ABA pathway [39]. In this case, the responses of *BpWRKY*s upon different times of ABA, salt, and cold treatment were explored using qRT-PCR. *BpWRKY*s showed expression specificity to different abiotic stresses. Commonly, the *BpWRKY*s responsive to ABA treatment overlapped with those responsive to salt treatment, including *BpWRKY3*, *BpWRKY33*, *BpWRKY40*, etc. This indicates that these genes are involved both in ABA signaling and salt stress. Meanwhile, there were BpWRKYs that were responsive to either treatment. For instance, the relative expression level of *BpWRKY61* was increased with salt treatment but was not significantly changed with ABA treatment, which indicates that this gene may be involved in salt stress response but not ABA signaling. Not as many cold-responsive *BpWRKY*s were as responsive to ABA or salt stress, which indicates that WRKY TFs are possibly more involved in ABA and salt stress than in cold stress. Furthermore, *BpWRKY17* itself was responsive to cold treatment and its encoding protein could interact with a cold-responsive TF, BpCBF7, which provides a new perspective on the mechanisms of BpWRKYs in response to cold stress. This study provides a reference for research on the roles of WRKY TFs in plant stress response. It offers insight into research on plant stress response mechanisms and the selection of gene resources for stress-tolerant tree breeding. Nevertheless, further experimental tools, such as gene overexpression and silencing, will be needed to more exquisitely identify the functions of these TFs. More attention needs to be paid to how WRKY TFs affect the stress response and tolerance of plants and the molecular mechanisms involved. It would be meaningful to use bioinformatics and molecular tools to identify more roles of WRKY TFs in plant stress response.

## 4. Materials and Methods

### 4.1. Identification and Characterization of WRKY Transcription Factors in Betula platyphylla

To obtain the genome information of *Betula platyphylla*, we searched and downloaded the genome, coding sequences (CDSs), and protein data of *Betula platyphylla* from the Phytozome v13.1 database (https://phytozome.jgi.doe.gov/pz/portal.html (accessed on 30 April 2023)). A BLASTP search [40] was performed and the Hidden Markov Model (HMM) file (PF03106) was retrieved from the Pfam protein family database (http://pfam.xfam.org/ (accessed on 25 June 2023)), and the putative BpWRKY family members were identified by using HMMER v3.1 [41]. Afterwards, the sequences of these putative members were further checked to determine if the conserved WRKY domain existed through InterPro (http://www.ebi.ac.uk/interpro/ (accessed on 29 June 2023)) and CDD (https://www.ncbi.nlm.nih.gov/Structure/cdd/wrpsb.cgi (accessed on 29 June 2023)). Finally, the putative BpWRKY family members were manually checked and classified according to the number and characteristics of their WRKY and zinc-finger motifs.

The information of the locations on the chromosomes of the confirmed *BpWRKY* genes was extracted from the phytozome database. In addition, these genes were named according to their location order on the chromosomes. These results were visualized using Tbtools-II (v1.120).

The basic characteristics of BpWRKYs, including the length of the genomic sequence, the number of amino acids, molecular weight, isoelectric point, aliphatic index and hydrophilic mean (GRAVY) score, chromosome location, and prediction of cellular localization were analyzed using ExPASy (http://www.expasy.org/ (accessed on 1 July 2023)).

### 4.2. Phylogenetic Analysis

The amino acid sequences of the WRKYs in *Arabidopsis thaliana* were downloaded from the phytozome database. The phylogenetic tree (1000 bootstrap replications) was constructed with the WRKY members in *Betula platyphylla* and *Arabidopsis thaliana* using the neighbor-joining (NJ) method via MEGA v7.0 software.

### 4.3. Gene Structure Analysis of BpWRKYs

The exon-intron structures were analyzed and visualized using the online software GSDS 2.0 (Gene Structure Display Server: http://gsds.cbi.pku.edu.cn/ (accessed on 5 July 2023)). The conserved motifs were analyzed using online tool MEME 5.0 (http://meme-suite.org/ (accessed on 5 July 2023)).

### 4.4. Gene Synteny Analyses of BpWRKYs

Synteny analysis was performed using Tbtools-II (v1.120) among *BpWRKY*s, as well as between *WRKY*s in *Arabidopsis thaliana* and *Betula platyphylla*. Tbtools-II (v1.120) was also used to calculate the non-synonymous (Ka) and synonymous (Ks) ratio for duplicated gene pairs.

### 4.5. Analysis of Cis-Acting Elements in the Promoters of BpWRKYs

A length of 2000 bp upstream sequence of each *BpWRKY* was extracted from the phytozome database (https: www.Phytozome.net, (accessed on 8 July 2023)). The *cis*-acting elements in the promoters of *BpWRKY*s were searched and analyzed using the online tool PlantCARE (http://bioinformatics.psb.ugent.be/webtools/plantcare/html/ (accessed on 8 July 2023)). Tbtools-II (v1.120) was used for visualization.

### 4.6. Plant Materials and Treatments

The plant materials used in this study were white birch (*Betula platytphylla*), preserved by the State Key Laboratory of Tree Genetics and Breeding, Northeast Forestry University (Harbin). They were grown in a greenhouse set with 16 h light/8 h darkness for a photoperiod at 25 °C. For tissue-specific analysis, the seedlings were grown in hydroponic culture (1/2 MS medium with 25 g/L sucrose, 0.02 mg/L NAA, and 0.4 mg IBA) for 60 d. After homogenization, plant samples including the root, stem, and leaf were collected, immediately frozen in liquid nitrogen, and preserved in a freezer at −80 °C for further experimental use.

The seedlings for ABA or salt (NaCl) treatment were also grown in hydroponic culture for 60 d. For ABA treatment, we adjusted the concentration of ABA in the hydroponic culture to 100 µM [42]. For salt treatment, we adjusted the concentration of NaCl in the hydroponic culture to 200 mM [43]. Plant samples (root) were collected according to the result of tissue-specific analysis after 0 h, 3 h, 6 h, 12 h, and 24 h of each treatment. For cold treatment (4 °C for 3 h) [44], the seedlings were first grown in hydroponic culture for 60 d and then transplanted to soil for another 30 days of growth. After the cold treatment, plant samples (root) were collected. Before sample preservation, all of the collected roots were rinsed with deionized water to remove nutrient solution and then the excess water was absorbed with absorbent tissue. These processes all had three biological replicates.

### 4.7. Data Processing and Expression Analysis

The total RNA of the plant samples was extracted using the Mega Pure Plant RNA Kit (Msunflowers Biotech Co., Ltd, Beijing, China). RNA quality was examined by gel electrophoresis and the concentration was measured. Afterwards, cDNA was synthesized using a reverse transcription kit (PrimeScriptTM RT reagent Kit, Takara Bio, Kusatsu, Japan). The primers (Appendix A) used in qRT-PCR were designed according to the downloaded full length cDNA sequences of *BpWRKY*s, and we chose *18S ribosomal RNA* as the internal reference gene [45]. qRT-PCR was performed using the THUNDERBIRD Next SYBR qPCR Mix (TOYOBO, Osaka, Japan). The reaction system was 20 µL and the conditions were as follows: pre-denaturation at 94 °C for 30 s, denaturation at 94 °C for 5 s, renaturation at 58 °C for 15 s, extension at 72 °C for 10 s, steps 2 to 4 for 45 cycles, melt curve for 6 s. The reaction was performed on the Applied Biosystems 7500 Fast Real-Time PCR System. Each reaction had three biological replicates. The relative expression levels of *BpWRKY*s were calculated using the 2^−∆∆Ct^ method [46]. Significant difference analysis was carried out by one-way ANOVA followed by Duncan’s multiple range test.

### 4.8. Yeast Two Hybrid (Y2H) Assay

According to the results of our previous study [36], we removed the transcriptional activating activity of BpCBF7 by cloning part (1 to 438 bp) of this gene, and introduced it into the pGBKT7 vector. The full-length CDS of *BpWRKY17* was introduced into the pGADT7 vector. These two recombinant vectors (pGBKT7-BpCBF7 (1–146 aa) and pGADT7-BpWRKY17) were jointly transformed into yeast competent cells. The yeast transformed with pGBKT-p53/pGADT7-T was the positive control, and the yeast transformed with pGBKTT7-LAM/pGADT7-T was the negative control. The yeast cells transformed with the combinations above were all cultured on nutrition-deprived yeast culture medium (TDO: SD/-Trp/-Leu/-His; QDO: SD/-Trp/-Leu/-His/-Ade) for 3–5 d at 30 °C. As a chromogenic substrate of the yeast GAL4 system, X-α-Gal was adjusted to 40 mg/L in culture medium, if added. AbA (Aureobasidin A), also as a symbol of activation of the yeast GAL4 system, was adjusted to 125 ng/mL in culture medium, if added.

### 4.9. Subcellular Localization

BpWRKY17 and BpWRKY67 were chosen for subcellular localization analysis because BpWRKY17 was studied in the Y2H assay and its encoding gene was a representative of the predicted chromosome-located *BpWRKY*s, while *BpWRKY67* was a representative of the predicted scaffold-located *BpWRKY*s. These two genes were combined with the pFGC-eGFP vector, respectively. The recombinant vectors and 35S-eGFP (positive control) were transformed, respectively, into Agrobacterium GV3101 using the liquid nitrogen freeze–thaw method. Then, transient transformation was performed to the leaves of one-month-old *Nicotiana benthamiana* seedlings by injection of these bacterial fluids. The transformed leaves were incubated in the dark for 48 h and then observed under a laser confocal microscope. Where the fluorescence could be detected indicated the location of fusion proteins in a cell.

## 5. Conclusions

In this study, a total of 68 WRKY TF family members were identified in *Betula platyphylla*. These members could be classified into four main groups according to the number and characteristics of their WRKY and zinc-finger motifs. Their basic physiochemical properties and phylogenetic relationships were analyzed. Different members exhibited different motif patterns and gene structures. Synteny analysis was performed among *BpWRKY*s as well as between *BpWRKY*s and *AtWRKY*s. The analysis of the *cis*-acting elements in the promoters of *BpWRKY*s showed that the elements were enriched in hormone response, growth and development, binding sites, and especially, stress response. In this case, after finding that most BpWRKYs were most expressed in the root, the responses of these TFs upon ABA, salt (NaCl), and cold treatment were investigated with birch root as the material. These TFs exhibited different transcriptional responses and response times towards different stress treatments. In addition, BpWRKY17 could interact with a cold-responsive TF, BpCBF7. The result of localization of BpWRKY17 and BpWRKY67 in the nucleus validated the result of the previous prediction. These results provide a foundation for the identification of the roles of BpWRKYs in plant stress response.

## Figures and Tables

**Figure 1 ijms-24-15000-f001:**
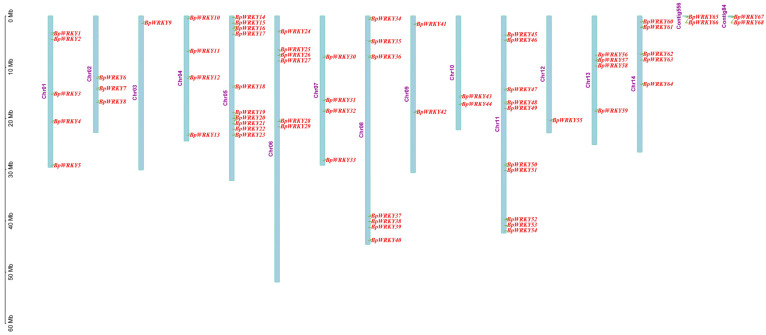
Chromosome location of *BpWRKY*s. The blue squares represent chromosomes. Chromosome numbers are shown on the left side. The scale bar on the left measures the chromosome length (Mb). Contig 556 and Contig 84 represent scaffolds.

**Figure 2 ijms-24-15000-f002:**
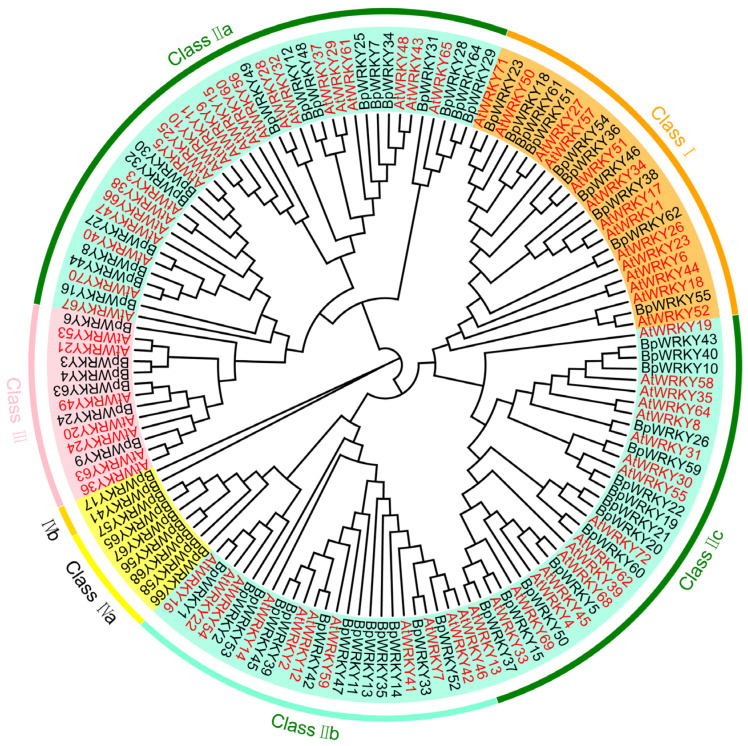
Phylogenetic analysis of WRKYs in *Betula platyphylla* and *Arabidopsis thaliana*. The phylogenetic tree (1000 bootstrap replicates) was constructed using MEGA 7.0. Classes with different colors represent different groups/subgroups. The red font indicates AtWRKYs and the black font indicates BpWRKYs.

**Figure 3 ijms-24-15000-f003:**
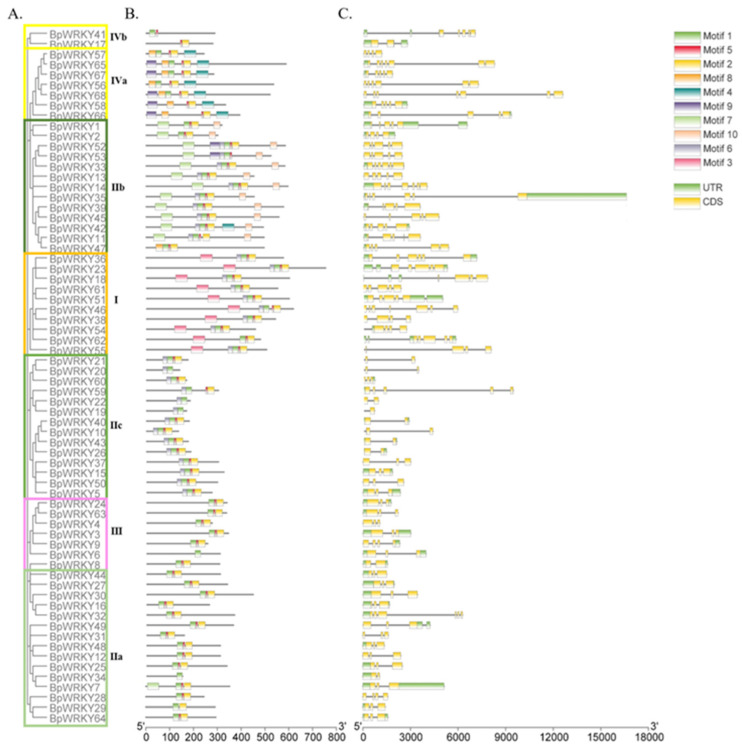
Exon–intron gene structure and structure of conserved motifs in BpWRKYs based on phylogenetic relationships. (**A**) Unrooted phylogenetic tree of the BpWRKYs. Different colored boxes indicate different groups. (**B**) Conserved motifs. Different colored boxes symbolize Motifs 1–10. (**C**) Exon–intron structure of BpWRKYs. Yellow boxes symbolize CDSs. Green boxes symbolize UTRs. Black lines symbolize introns. The scale bars measure the length (bp) of the gene.

**Figure 4 ijms-24-15000-f004:**
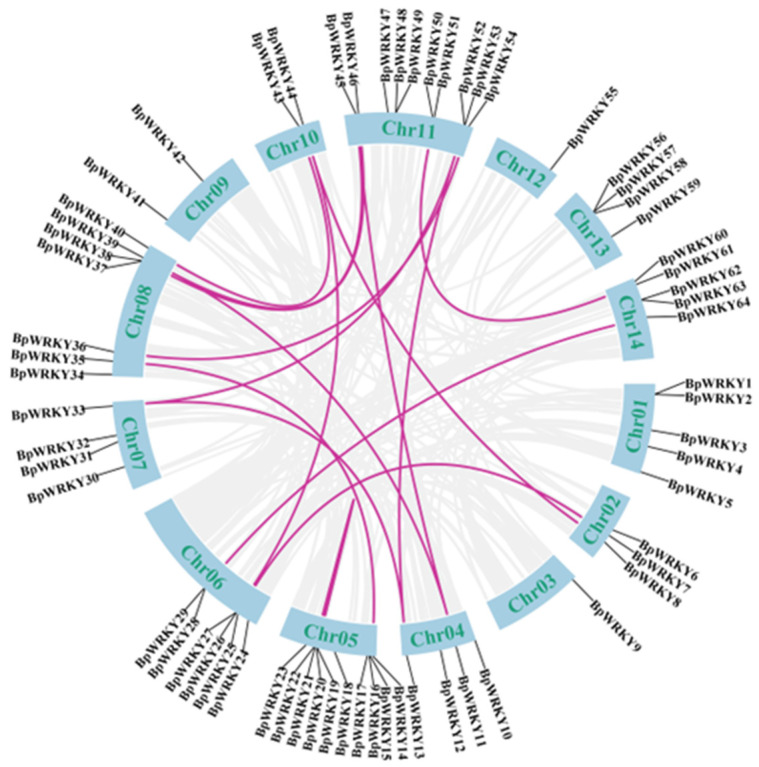
Synteny analysis of *BpWRKY*s. The gray lines represent all of the collinear blocks in the *Betula platyphylla* genome. The pink lines represent syntenic *BpWRKY* gene pairs. The chromosome number is shown in green on each chromosome.

**Figure 5 ijms-24-15000-f005:**
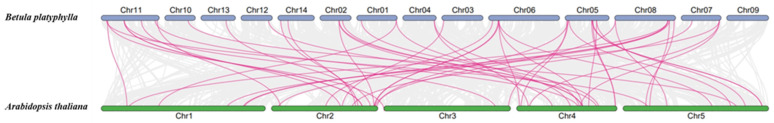
Synteny analysis of WRKYs in *Betula platyphylla* and Arabidopsis thaliana. The grey lines in the background represent collinear blocks in *Betula platyphylla* and Arabidopsis thaliana, respectively. The pink lines represent syntenic WRKY gene pairs. The numbers on the chromosomes indicate chromosome numbers.

**Figure 6 ijms-24-15000-f006:**
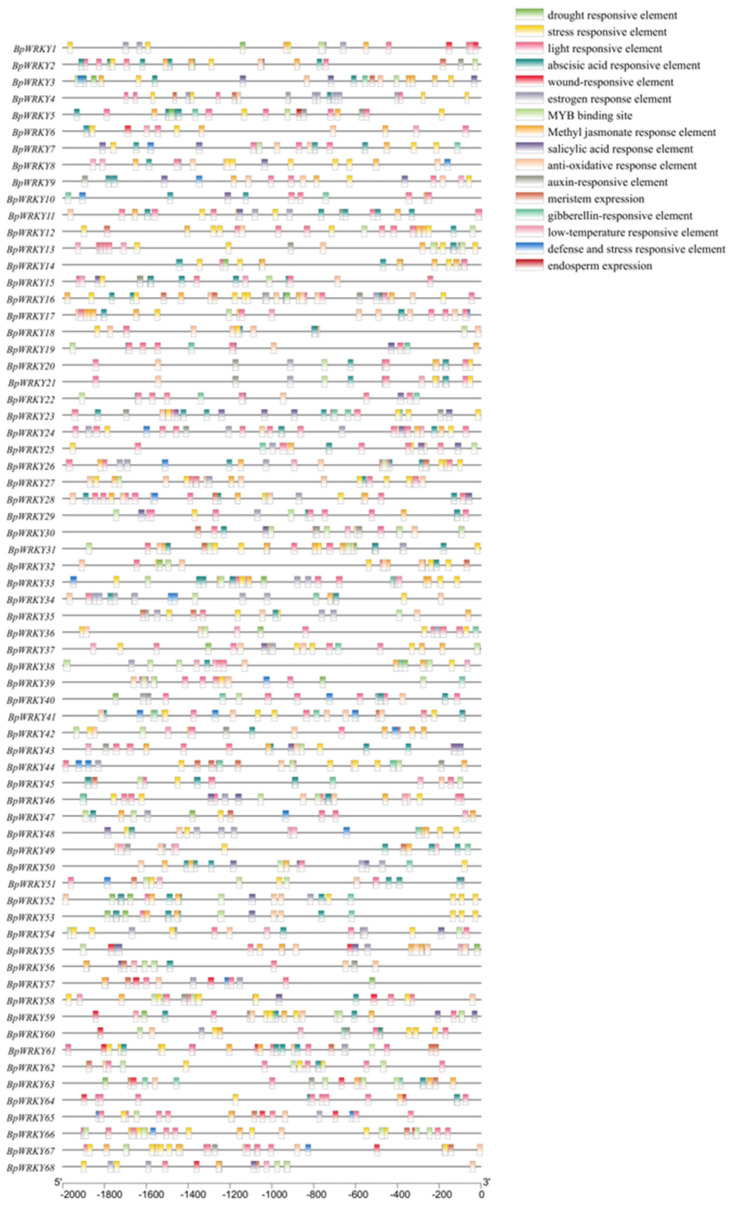
*Cis*-acting element analysis of the promoters of *BpWRKY*s. Different colored boxes represent different *cis*-acting elements.

**Figure 7 ijms-24-15000-f007:**
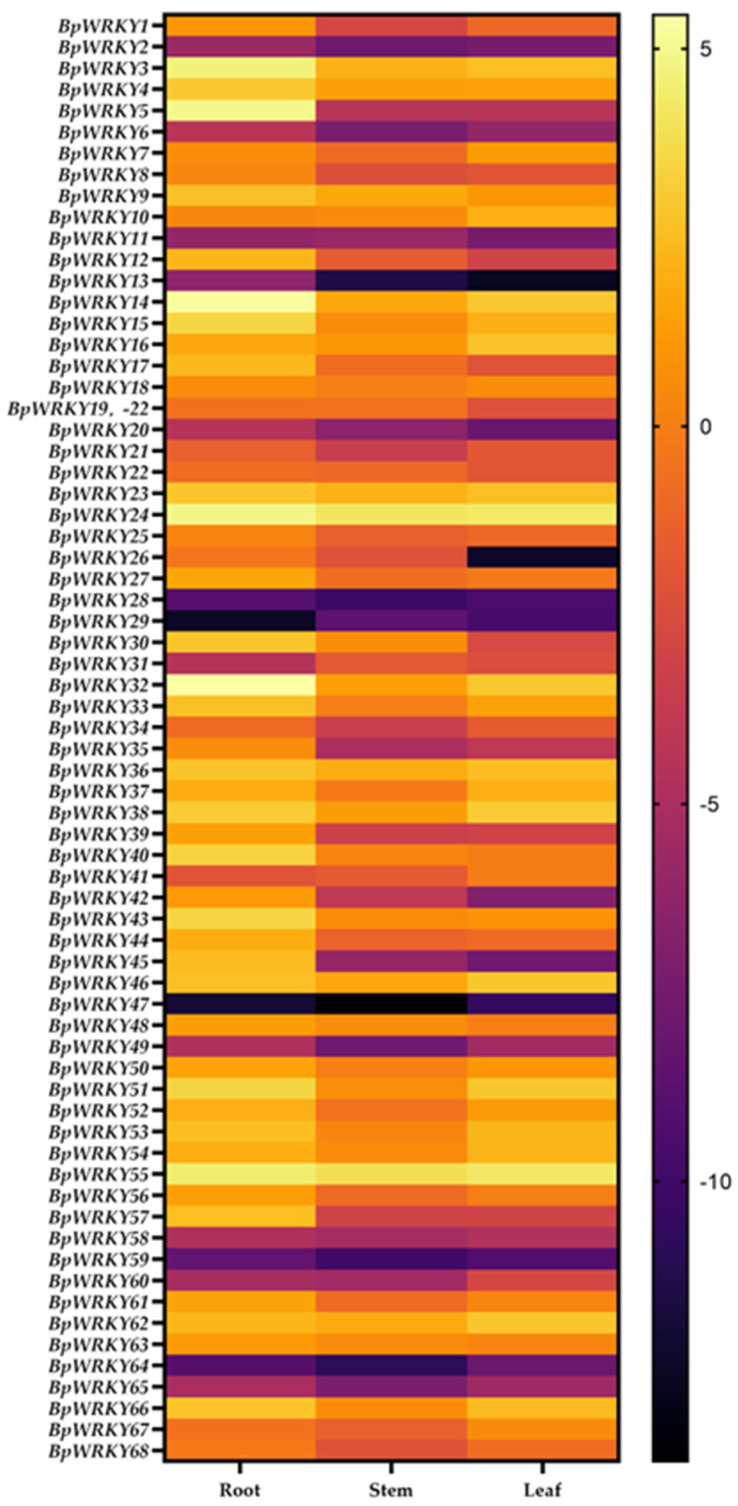
Tissue-specific expression analysis of *BpWRKY*s. Tissue-specific expression analysis is performed with the root, stem, and leaf of birch. The 2^−ΔΔCt^ method was used to calculate the transcription levels of *BpWRKY*s and the log_2_ value of each *BpWRKY* was used to show its relative expression level. The scale bar is shown on the right side of the heatmap.

**Figure 8 ijms-24-15000-f008:**
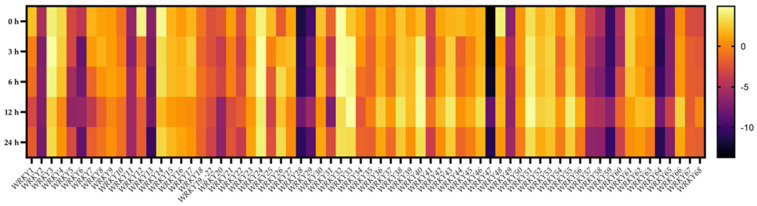
Expression patterns of *BpWRKY*s under ABA treatment. The concentration of ABA treatment is 100 μM and the treatment times are 0 h, 3 h, 6 h, 12 h, and 24 h. The 2^−ΔΔCt^ method was used to calculate the transcription levels of *BpWRKYs* and the log_2_ value of each *BpWRKYs* was used to show its relative expression level. The scale bar is shown on the right side of the heatmap.

**Figure 9 ijms-24-15000-f009:**
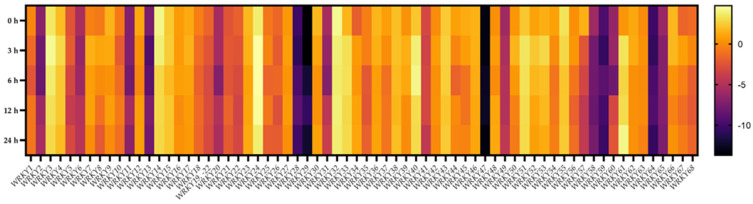
Expression patterns of *BpWRKY*s under salt treatment. The concentration of salt treatment is 200 mM, and the treatment times are 0 h, 3 h, 6 h, 12 h, and 24 h. The 2^−ΔΔCt^ method was used to calculate the transcription levels of *BpWRKYs* and the log_2_ value of each *BpWRKYs* was used to show its relative expression level. The scale bar is shown on the right side of the heatmap.

**Figure 10 ijms-24-15000-f010:**
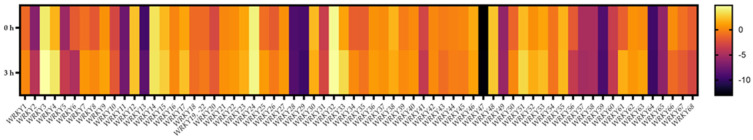
Expression patterns of *BpWRKY*s under cold treatment. The treatment condition was 4 °C for 3 h. The 2^−ΔΔCt^ method was used to calculate the transcription levels of *BpWRKYs* and the log_2_ value of each *BpWRKYs* was used to show its relative expression level. The scale bar is shown on the right side of the heatmap.

**Figure 11 ijms-24-15000-f011:**
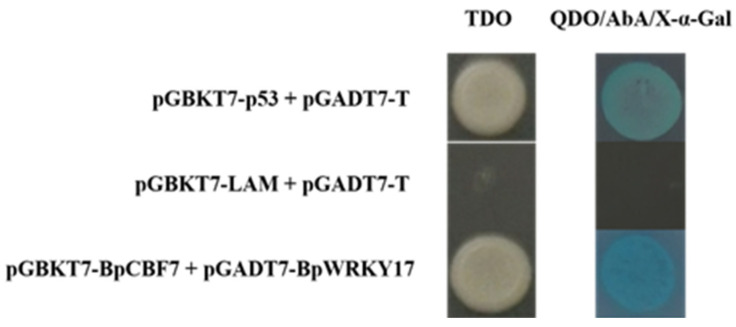
Interaction between BpWRKY17 and BpCBF7 validated by yeast 2 hybrid assay. pGBKT7-p53/pGADT7-T is the positive control. pGBKT7-LAM/pGADT7-T is the negative control. AbA concentration in the nutrition-deprived medium was 125 ng/mL. X-α-Gal concentration in the nutrition-deprived medium was 40 mg/L.

**Figure 12 ijms-24-15000-f012:**
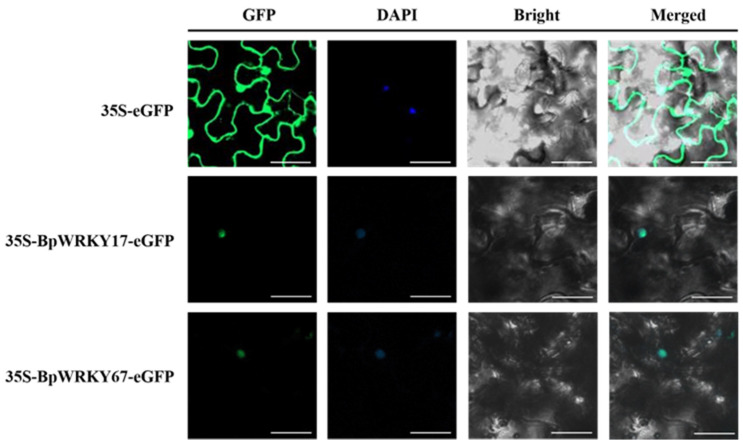
Subcellular localization of BpWRKY17 and BpWRKY67. The control (35S-eGFP) and fusion vectors (35S-BpWRKY17-eGFP, 35S-BpWRKY67-eGFP) were separately expressed in tobacco (*Nicotiana benthamiana*) leaves. GFP indicates green fluorescent protein signal. DAPI acts as a marker of nuclear localization. Bright indicates bright field. Merged indicates the merged signal. Scale bar = 10 μm.

**Table 1 ijms-24-15000-t001:** The Ka/Ks ratios of syntenic *BpWRKY* gene pairs.

Duplicated Gene Pairs	Ka	Ks	Ka/Ks	Effective Length * (bp)
*BpWRKY19/BpWRKY22*	0	0	--	501
*BpWRKY20/BpWRKY21*	0.064	0.100	0.636	366
*BpWRKY7/BpWRKY25*	0.407	1.767	0.230	963
*BpWRKY13/BpWRKY33*	0.404	2.425	0.167	1260
*BpWRKY11/BpWRKY39*	0.335	1.266	0.265	1335
*BpWRKY14/BpWRKY35*	0.424	1.507	0.282	1227
*BpWRKY40/BpWRKY43*	0.294	1.287	0.229	510
*BpWRKY8/BpWRKY44*	0.458	3.314	0.138	699
*BpWRKY27/BpWRKY44*	0.363	2.688	0.135	678
*BpWRKY11/BpWRKY45*	0.370	1.534	0.241	1362
*BpWRKY39/BpWRKY45*	0.392	1.401	0.280	1533
*BpWRKY38/BpWRKY46*	0.260	1.086	0.239	1425
*BpWRKY36/BpWRKY54*	0.399	1.103	0.362	1080
*BpWRKY13/BpWRKY52*	0.385	2.101	0.183	1299
*BpWRKY33/BpWRKY52*	0.267	2.081	0.128	1578
*BpWRKY51/BpWRKY61*	0.307	1.643	0.187	1614
*BpWRKY29/BpWRKY64*	0.012	0.081	0.150	864

* Note: Effective length represents the length of a homologous fragment.

## Data Availability

Not applicable.

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
