# Peer review of "Genome-Wide Identification and Characterization of WRKY Transcription Factors in Betula platyphylla Suk. and Their Responses to Abiotic Stresses"

_ijms, 2023, doi:10.3390/ijms241915000_

Round 1

Reviewer 1 Report

The manuscript entitled ‘Genome-wide Identification and Characterization of WRKY Transcription Factors in Betula Platyphylla and their Responses to Abiotic Stresses’ describes the identification and molecular characterization of 68 members of WRKY transcription factors from white birch. They were phylogenetically classified based on conserved domains, characterized in silico for genomic location, gene synteny analysis, promoter analysis, biochemical features of the predicted proteins and subcellular localization. The differential expression was studied for tissue specificity and induction with respect to abiotic stresses (salt, cold) and ABA hormone induction. The nuclear localization of two members were experimentally verified and a crosstalk of WRKY 17 and CBF7 proteins was also revealed. Overall, the study is comprehensive in terms of bioinformatic and in planta analysis of role of WRKY transcription factors in plant development and its interplay with other transcriptional regulators mediating tolerance to abiotic stresses.

Suggested revisions: 

1.    In lines 53, 138, 156, 348, 349 : Group VI or IV? It is noted in the figures as IV.

2.    Line 305-306: ‘pGBKT7-p53/pGADT7-T is the negative control.’ Is this correct? Section 4.8 explains correctly.

Reviewer 2 Report

Th discussion is more a summary than a discussion of present results comparing the state of the art of previous papers.

In Rows 83- 85 and 327 are listed a series of species but these are incomplete. For example Durum wheat is missing (please see DOI: 10.1089/omi.2011.0081 and references therein.

A clear statement indicating the manuscript objectives is missing.

Species names should be in italic with general initial in capital while the species initial should be in small letter.

The size of the text in all figures is too small.

Row 167 please delete "of the 68 BpWRKYs"

Table 2 caps delete one of the two "of"

In Table 2, is it necessary to have a precision at the 9 decimal number?

Row 242 please delete "And". Please, also in other sentence try to avoid to start a sentence with "And".

Language is almost fine, at part the sentences started with "And"

Reviewer 3 Report

The manuscript “Genome-wide Identification and Characterization of WRKY Transcription Factors in Betula Platyphylla and their Responses to Abiotic Stresses” aimed to identify and describe the WRKY TFs family in Betula platyphylla by using both bioinformatics tools and tissue-specific expression analysis. Due to the lack of information on WRKTY TFs in the studied species and their relevance in stress response, I believe that the presented results are worth publishing in IJMS and will be of interest to a wide range of its readers.

My overall impression after reading the manuscript is positive. I find the conclusions and data interpretation valid and reliable. The whole manuscript is written in understandable language but in my opinion, several issues need to be presented in more detail or clarified (see below). The information provided in the M&M is sufficient. However, some figures could be of better quality to improve their readability (see below).

Here I list my suggestions about how to improve the manuscript:

Introduction:

-     Lines 54-80: This paragraph gives a lot of information about WRKY TFs family in many plant species. However, little is written about woody species.

-      Lines 81-86: I believe it would be better if these lines were moved before the second paragraph (before lines 54-80).

-       There should be a full scientific name of Betula platyphylla in the title and at first mention in the abstract and in the introduction. Also, why is platyphylla capitalized throughout the manuscript?

-  There should be more information about Betula platyphylla itself – importance, typical habitat, ecological and economic importance, threats to the species/populations, etc.

Results:

-       Line 132: five groups – should be changed to “four”?

-    Paragraph 2.6 should be rewritten (paragraph 2.7 is better, so it could serve as an example). It should focus only on the most differentiating examples and the overall pattern (there seems to be that in most cases the higher expression is present in the root, but see for example BpWRKY29 or 31). The authors may also provide ranges of relative expression levels (separately for root, stem, and leaf). Also, in line 223 there is information about 3722 higher expression level – it is not proportional to the scale presented in Fig. 7.

-  Paragraph 2.10 – I am wondering what was the reason to choose BpWRKY17 for this particular analysis.

Discussion:

-       It can be seen in Fig. 2 that classes IVa and IVb have no related genes in A. thaliana. Can the authors make any comment on this finding? Does this class play any distinctive role?

-  I think there should be some additional comment on the most distinguished findings of tissue-specific expression analysis presented in Fig. 7 (e.g. BpWRKY26).

Figures:

-       Figure 3: Could the authors add “bp” to the axis?

-    If possible, the quality of Figures 3 and 6 should be improved to make them more readable.

-       Figure 7: the scale is not proportional to the values in paragraph 2.6.

-     Figure 11 (caption): It is written that pGBKT7-p53/pGADT7 is the positive control. In fact, it is a negative control.

The English language requires only minor corrections:

-       line 196 (below Table 2): fragement à fragment

-       line 205: Methyl à methyl

-       line 219: was extracted and then through reverse transcription into cDNA… à was extracted and then reversely transcribed into cDNA…

Reviewer 4 Report

Table 1 can be moved to supplementary. Too many figures. Figures 5 and 6 can be in supplementary as well. 

·        Syntenic Analysis: While syntenic gene pairs are discussed, the study could delve deeper into the evolutionary implications of these syntenic relationships.

·        Comparison with Other Species: The study incorporates AtCBFs in the phylogenetic analysis but could also compare the identified BpWRKYs with WRKYs from other closely related species.

·        Regulatory Elements: The study mentions cis-acting elements but doesn't explore them in depth. Understanding these elements could provide insights into the gene regulation mechanisms.

·        Functional Divergence: Given that some BpWRKYs showed specificity in stress response, a discussion on potential functional divergence within this gene family could be enlightening.

·        Biological Significance: While the study provides a wealth of data, it could benefit from a discussion that ties these findings to broader biological or ecological significance.

·        Future Directions: The study could offer more explicit recommendations for future research, beyond the general call for gene overexpression and silencing studies.

English is fine

Round 2

Reviewer 4 Report

Accept

Accept